# Shiga Toxin-Producing *Escherichia coli* (STEC) Associated with Calf Mortality in Uruguay

**DOI:** 10.3390/microorganisms11071704

**Published:** 2023-06-29

**Authors:** Magalí Fernández, María Laura Casaux, Martín Fraga, Rafael Vignoli, Inés Bado, Pablo Zunino, Ana Umpiérrez

**Affiliations:** 1Departamento de Microbiología, Instituto de Investigaciones Biológicas Clemente Estable, Avenida Italia 3318, Montevideo 11600, Uruguay; mafertri@gmail.com (M.F.); pzunino@iibce.edu.uy (P.Z.); 2Plataforma de Investigación en Salud Animal, Instituto Nacional de Investigación Agropecuaria (INIA), Estación Experimental INIA La Estanzuela, Ruta 50, Km 11, Colonia 70000, Uruguay; lcasaux@inia.org.uy (M.L.C.); mfraga@inia.org.uy (M.F.); 3Departamento de Bacteriología y Virología, Facultad de Medicina, UdelaR, Av Alfredo Navarro 3051, Montevideo 11600, Uruguay; rvignoli@higiene.edu.uy (R.V.); ibado@higiene.edu.uy (I.B.)

**Keywords:** Shiga toxin-producing *E. coli*, dairy calf, mortality, MDR STEC

## Abstract

In Uruguay, the mortality of dairy calves due to infectious diseases is high. *Escherichia coli* is a natural inhabitant of the intestinal microbiota, but can cause several infections. The aim of the work was to characterize *E. coli* isolates from intestinal and extraintestinal origin of dead newborn calves. Using PCR, virulence gene characteristics of pathogenic *E. coli* were searched. The pathogenic *E. coli* were molecularly characterized and the phylogroup, serogroup and the Stx subtype were determined. Antibiotic susceptibility was determined using the Kirby–Bauer disk diffusion method and plasmid-mediated quinolone resistance (PMQR) genes with PCR. Finally, clonal relationships were inferred using PFGE. Gene characteristics of the Shiga toxin-producing *E. coli* (STEC), Enteropathogenic *E. coli* (EPEC) and Necrotoxigenic *E. coli* (NTEC) were identified. The prevalence of the *iucD*, *afa8E*, *f17*, *papC*, *stx1*, *eae* and *ehxA* genes was high and no *f5*, *f41*, *saa*, *sfaDE*, *cdtIV*, *lt*, *sta* or *stx2* were detected. The prevalence of STEC gene *stx1* in the dead calves stood out and was higher compared with previous studies conducted in live calves, and STEC LEE+ (Enterohemorrhagic *E. coli* (EHEC)) isolates with *stx1*/*eae*/*ehxA* genotypes were more frequently identified in the intestinal than in the extraintestinal environment. *E. coli* isolates were assigned to phylogroups A, B1, D and E, and some belonged to the O111 serogroup. *stx1a* and *stx1c* subtypes were determined in STEC. A high prevalence of multi-resistance among STEC and *qnrB* genes was determined. The PFGE showed a high diversity of pathogenic strains with similar genetic profiles. It can be speculated that EHEC (*stx1*/*eae/ehxA*) could play an important role in mortality. The *afa8E*, *f17G1* and *papC* genes could also have a role in calf mortality. Multidrug resistance defies disease treatment and increases the risk of death, while the potential transmissibility of genes to other species constitutes a threat to public health.

## 1. Introduction

*Escherichia coli* colonizes the intestines of animals a few hours after birth and is part of the normal microbiota of the digestive tract of warm-blooded animals, including cattle [1]. Nevertheless, some variants have acquired virulence attributes that allow them to cause disease in humans and animals [1,2,3,4].

Shiga toxin-producing *E. coli* (STEC) is considered a highly relevant virotype in public health, responsible for severe outbreaks in humans [5]. STEC infections in humans are generally associated with symptoms ranging from mild diarrhea to Hemorrhagic Colitis (HC), although, in severe cases, it can lead to kidney damage and neurological disorders [6,7,8]. STEC strains are characterized by the production of Shiga toxin (Stx) 1 and/or Stx2 toxins of at least 107 known variants, each with differences in their antigenicity and cellular toxicity [6,9]. The cell target of Stx depends on the number of specific receptors present on the cell membrane: Gb3 (Gala1-4Gal) for Stx1 and Stx2, and Gb4 (GalNAcb1-3Gala1-4Gal) for Stx2e and Stx2f. In piglets and humans, the main target cells are endothelial cells of small arteries, gastrointestinal mucosa, kidneys, the brain and other tissues, leading to hemorrhages with consequent tissue damage and clinical syndromes [6]. Calves lack Gb3 receptors in the endothelial cells of their blood vessels, which has led to the belief that they are intrinsically resistant and asymptomatic carriers of the virotype [7]. However, recent studies have provided evidence that there are different types of target cells for Stx in cattle, such as intestinal and peripheral lymphocytes, colonic epithelial cells and macrophage-like cells residing in the bovine mucosa [10]. Stx can affect the function of T and B lymphocytes, suggesting that they could modulate the immune response with possible consequences for STEC survival, colonization of the bovine gastrointestinal tract and establishment of a symbiotic relationship with the adult host [11]. Although various species, such as goats, sheep, bison, horses, pigs and water buffalo, are capable of harboring STEC, cattle are recognized as the main reservoir [12].

STEC isolates are divided into a group that contains the Locus of Enterocyte Effacement (LEE) pathogenicity island and another that does not [8]. LEE contains the gene encoding the adhesin intimin, necessary to cause intestinal lesions characteristic of attaching and effacing (A/E). STEC LEE+ (also known as Enterohemorrhagic *E. coli* EHEC) can have severe manifestations such as infectious colitis in infected hosts [8]. In contrast, STEC LEE− cannot cause A/E injury and, therefore, the molecular mechanisms of pathogenicity differ from those of EHEC [13].

Traditionally, the *E. coli* STEC O157:H7 serogroup has been recognized as responsible for important epidemiological outbreaks in humans [5,14]. However, more recently, other non-O157:H7 serogroups known as the “big six” (O26, O45, O103, O111, O121 and O145) have been identified. Their incidence has increased significantly in recent years and they are considered a risk to public health [15]. Like O157, these serogroups can be isolated from ruminant feces, intestinal content and from feed [6].

It is well-known that *E. coli* represents an important reservoir of resistance genes, many of which are transferred horizontally and may be responsible for treatment failures in both veterinary and human medicine. Antimicrobial resistance in *E. coli* is considered one of the main challenges worldwide. According to the WHO, this microorganism is considered a priority for antibiotic resistance research, whose study provides information on the antimicrobial abuse, transmission dynamics and development of new resistances [2,16,17].

Additionally, other virotypes are associated with enteric disease. Enterotoxigenic *E. coli* (ETEC) and Enteropathogenic *E. coli* (EPEC) have been an important cause of human diarrhea for decades, especially in children under five years old, and also can cause disease in animals, whereas Necrotoxigenic *E. coli* (NTEC) strains have been isolated from cases of enteritis in ruminants, pigs, rabbits, dogs and horses, and from extraintestinal infections in dogs, cats, humans and pigs [1,18,19,20].

In Uruguay, the production of milk and dairy products is one of the most important agricultural activities both for domestic consumption and for export. The significant demand for them has led to the intensification of this activity, circumstances that favor the transmission of pathogens [21]. It is known that the mortality rates of animals of perinatal age raised in dairy farms exceed those reported for Argentina, the United States and France and are mainly associated with infectious diseases such as neonatal calf diarrhea (NCD) [22].

The presence of pathogenic *E. coli* isolates in the feces of calves has been reported in Uruguayan dairy herds, in both healthy and diseased animals with NCD, and its virulence profile, zoonotic potential, resistance phenotype and presence of transferable resistance genes have been described [23,24,25,26]. Taking into account the previous works carried out in this category of bovine, so far the proportion of STEC in feces is low, but with an important proportion of potentially harmful non-O157 STEC isolates [24,25]. However, the study of STEC associated with calves’ mortality remains unknown.

The objective of this work was to characterize *E. coli* isolates recovered from the intestinal and extraintestinal environment of dead neonatal dairy calves in Uruguay; two hundred and twenty-one isolates were characterized according to their virulence profiles. A selection of 20 *E. coli* characterized as EPEC, STEC and NTEC were examined to identify their phylogroups, serogroups, Shiga toxin subtypes and antimicrobial resistance profiles. In addition, Pulsed Field Gel Electrophoresis (PFGE) was used for subtyping and determining the relatedness/diversity between these isolates.

## 2. Materials and Methods

Between 2015 and 2017, personnel from Plataforma de Salud Animal in INIA-La Estanzuela recovered and identified *E. coli* isolates from different biological samples of dead calves (Appendix A). The 17 dead animals that made up that study belonged to the departments of Colonia, San José, Canelones and Río Negro in Uruguay and presented signs of decay, weakness, pneumonia, septicemia or diarrhea before their death (Table 1). All studied animals came from dairy farms, except for one from an intensive meat production establishment.

Of each plate with lactose-positive colonies, at least one up to ten colonies were studied in our laboratory. A total of 221 *E. coli* isolates were recovered during necropsies.

The presence of virulence genes of ETEC (*lt* and *sta*), STEC (*stx1*, *stx2* and *saa*), EPEC (*eae*) and NTEC (*cnf1* and *cnf2*) were evaluated using PCR. Likewise, genes coding for adhesins (*afa8E* and *clpG*), fimbriae (*f5*, *f41*, *f17A*, *f17GI*, *f17GII*, *papC* and *sfaDE*), toxins (*ehxA*, *cdtIII* and *cdtIV*) and the siderophore aerobactin (*iucD*) were searched in the intestinal and extraintestinal pathogenic *E. coli* strains. Primers for the partial detection of each gene were taken from the literature and the PCR protocols were previously developed in our laboratory [23,26].

A selection of isolates was carried out considering that samples with 2 or more different genetic profiles provide an isolate for each particular genotype, regardless of the number of positive isolates for each virulence profile [27]. All the isolates that exhibited unique genetic profiles in the biological sample of the dead animal to which they belonged and that could be categorized within one of the STEC, EPEC, ETEC and/or NTEC virotypes were selected. This group (n = 20) underwent further molecular and phenotypic characterization. In addition, molecular typing using PFGE and antibiotic susceptibility testing was performed to the selected isolates.

PCRs were performed to amplify gene sequences encoding O-antigenic regions corresponding to serogroups O157, O26, O45, O103, O111, O121 and O145 [28]. Additionally, multiplex PCRs were performed to assign isolates to groups A, B1, B2, C, D, E, F and Clade I [29]. The presence of genes encoding the Stx1 subtypes was evaluated according to previously established protocols [30].

To assess susceptibility to antimicrobials, the Kirby–Bauer disc-diffusion method was used [31]. For each selected isolate, 14 antibiotics were tested: ampicillin (AMP), ceftazidime (CAZ), ceftriaxone (CRO), amoxicillin-clavulanic acid (AMC), cefuroxime (CXM), cefepime (FEP), nalidixic acid (NAL), enrofloxacin (ENR), ciprofloxacin (CIP), amikacin (AK), gentamicin (CN), tobramycin (TOB), streptomycin (STR) and trimethoprim-sulfamethoxazole (SXT). All antibiotic discs were purchased from Oxoid. Quality control was performed using ATCC *E. coli* 25922. The interpretation of the results was performed according to CLSI 2019, except for ENR, which was interpreted using Veterinary Antimicrobial Susceptibility Testing [31]. Additionally, using simple PCR, we detected the presence of Plasmid-Mediated Quinolone Resistance genes (PMQR) via partial amplification of the *qnrA*, *qnrB*, *qnrC*, *qnrD*, *qnrE*, *qnrVC* and *aac(6′)Ib-cr* genes as previously described, and the identity was confirmed using direct sequencing [24].

Clonality was determined using the PFGE technique according to the protocol established by PulseNet [32]. *Salmonella enterica* subsp. *enterica* serovar Braenderup H9812 was used as a control strain. Restriction profiles were analyzed with the GelCompar II bioinformatics program (Applied Maths, Version 6.5). Dendrograms were constructed using the UPGMA method and the Dice coefficient, with an optimization of 1% and a tolerance of 1%. Strains whose restriction profiles had a similarity coefficient greater than or equal to 85% were considered genetically related, while those with a coefficient of 100% were considered clones [33].

In order to evaluate the association between the presence of the virulence genes tested and the origin of the isolates (intestinal or extraintestinal), the Odds Ratio (OR) test was used, considering statistical significance when the *p*-value was less than 0.05 [34].

## 3. Results

### 3.1. Detection and Identification of Virulence Genes in E. coli

We studied the presence of 21 *E. coli* virulence genes in 221 isolates belonging to 17 dead calves. Fifteen animals (88%) presented at least one isolate carrying some of the studied genes. No virulence genes were detected in *E. coli* isolates from the urine, kidney, heart or cerebrospinal fluid (Appendix A).

In 9/17 dead calves, we were able to detect *E. coli* virotypes, highlighting the prevalence of STEC (6/17 animals), followed by EPEC and NTEC (2 and 1 animal, respectively) (Appendix A). The *iucD* gene was the most frequently identified among the animals, amplifying in 82% of the calves (14/17 animals). The observed frequency for *f17A* and *papC* was 59% (10/17 animals), for *afa8E* 53% (9/17 animals), for *ehxA* 47% (8/17 animals), for *eae* 41% (7/17 animals), for *stx1* 35% (6/17 animals), for *clpG* 24% (4/17 animals), for *f17GI* and *f17GII* 18% (3/17 animals), for *cdtIII* and *cnf1* 12% (2/17 animals) and for *cnf2* 6% (1/17 animals). The genes *f5*, *f41*, *saa*, *sfaDE*, *cdtIV*, *lt*, *sta* and *stx2* were not detected in any calf (Figure 1 and Appendix A). Taking into account the presence of virulence genes per animal, it was determined that there was an association in the co-occurrence of the *stx1*, *eae* and *ehxA* genes (OR = 44; 95% CI, 2.90–667.17).

### 3.2. Occurrence and Diversity of Virulence Genes in Intestinal and Extraintestinal Isolates

According to their origin, the isolates were grouped as intestinal or extraintestinal. The intestinal isolates included those recovered from the intestine and feces of dead calves, while the extraintestinal group included isolates from the brain, brain spinal fluid, kidney, urine, heart, lung, liver, mesenteric lymph nodes (MLN), spleen and bladder (Appendix A).

All the evaluated genes were detected in both environments, except for *cnf1* and *cnf2* genes, which were only detected in intestinal isolates (Figure 2).

The prevalence in the intestinal and extraintestinal environment varied for each gene (Figure 2). Such differences were not significant for any gene, except for *stx1*, *eae* and *ehxA*. Those were significantly more represented in the intestinal than in the extraintestinal isolates (OR = 8.25; 95% CI, 1.15–59.01; OR = 12.83; 95% CI, 1. 69–97.20 and OR = 7.78; 95% CI, 1.20–50.43, respectively) (Figure 2).

### 3.3. Molecular Characterization of EPEC, STEC and NTEC

Isolates with unique virulence profiles in each biological sample having EPEC, STEC and NTEC genes were selected (n = 20) for further characterization (Table 2). Of the selected isolates, 75% belonged to the STEC virotype (70% EHEC and 5% STEC LEE−), 10% to hybrid EHEC/NTEC, 10% were EPEC and 5% were NTEC.

#### 3.3.1. Stx1 Typing

A total of 17 *stx1*+ isolates were investigated using PCR, from which 16 resulted in a *stx1*a/*stx1*c Shiga toxin subtype (Table 2). Isolate 15.40 was negative for all the studied Stx1 variants.

#### 3.3.2. Phylogenetic Group Assignment

The selected isolates were assigned to phylogroups A, B1, C and E. The most prevalent group among the isolates was B1, being identified in eight isolates (40%), group E was identified in six isolates (12%) and groups A and C were each represented with three isolates (6%) (Table 2). Isolates recovered from the extraintestinal environment were grouped into phylogroups B1 and C, and the intestinal ones were assigned to phylogroups A, B1 and E. STEC was grouped in B1, C and E; EPEC in E; NTEC in B1; and the NTEC/STEC hybrid isolates belonged to group A (Table 2).

#### 3.3.3. Serogroup Determination

Using the PCR technique, partial gene sequences encoding O157, O26, O45, O103, O111, O121 and O145 O-antigenic regions were amplified. Nine STEC isolates (45%) were characterized as serogroup O111, while the rest of the isolates could not be assigned to any of the tested serogroups (Table 2).

### 3.4. Antibiotic Susceptibility Profile

All the isolates were resistant to AMP (100%), whereas 19 showed resistance to STR (95%), 17 to CIP and NAL (85%), 13 to ENR (65%), 6 to CN (30%) and 5 to SXT (25%), 4 to TOB (20%) and 3 isolates were resistant to AMC and CXM (15%) (Table 3). All the isolates presented resistance of at least three antibiotics (up to nine different antimicrobials) and 16 were considered MDR, that is, they exhibited resistance to at least three categories of antibiotics [35]. These isolates were, at least, resistant to AMP, STR and some of the quinolones tested (ENR, CIP and/or NAL) simultaneously. Resistance to CAZ, FEP, CRO and AK was not detected (Table 3). None of the isolates produced Extended-Spectrum Beta-Lactamases (ESBL).

#### Presence of PMQR Genes in STEC and EPEC Isolates

A subgroup of 16 isolates resistant to CIP and NAL was tested for the presence of PMQR genes. The *qnrB* gene was detected in 11 isolates from dead calves (8 EHEC, 2 EPEC and 1 NTEC isolate), and the *qnrB19* variant was confirmed in 3 (Table 3).

### 3.5. Clonal Relationship of STEC/EPEC/NTEC Related to Calves’ Mortality

The intraspecific diversity of the isolates was determined by analyzing the restriction profiles using the GelCompar II software. Isolate 10.4, belonging to the lung of animal n°10, was not typed by PFGE. The 19 typified isolates were classified into 13 unique isolates grouped into 6 genetic variants. The variants were integrated by up to three unique isolates. In the variants, *E. coli* assigned to different virotypes and recovered from different biological samples, animals and herds were recognized (Figure 3).

In addition, 12 restriction profiles were aligned in five groups of clones, each with isolates of identical restriction profiles (100% similarity coefficient). In some cases, the isolates had been recovered from different animals (isolates 9.5–10.2 and 10.6–11.2) even though they belonged to the same farm (Figure 3).

## 4. Discussion

The mortality rate of neonatal calves in Uruguay is one of the highest in America, with NCD as a common cause of death [22]. Although most strains have a commensal relationship in the gastrointestinal tract, *E. coli* has been widely identified as responsible for septicemia and neonatal mortality in calves [2,36]. This study aimed to characterize for the first time intestinal and extraintestinal *E. coli* isolates from dead newborn calves recovered between 2015 and 2017 in Uruguay.

The characterization of 221 *E. coli* isolates from 17 dead calves revealed the presence of 13 of the 21 virulence genes tested.

The prevalence of *stx1*, *eae* and *ehxA* genes were 35%, 41% and 47%, respectively, extremely higher values compared with previous works on alive calves in Uruguay (OR = 11.85; 95% CI, 4.19–33.52, OR = 9.75; 95% CI, 4.05–23.49 and OR = 18.02; 95% CI, 6.65–48.84, respectively) and Argentina (OR = 18.93; 95% CI, 5.39–66.53, OR = 3.97; 95% CI, 2.01–7.82, respectively) [23,26,27]. These genes were found associated in the animals (OR = 44; 95% CI, 2.90–667.17), with a higher representation in the intestinal environment (Figure 2). The same genotype (*stx1*/*eae*/*ehxA*) has been reported in a higher frequency than other gene combinations in diarrheic calves in Iran [37]. Unlike the cytotoxicity observed in humans, in bovines, Stx toxins act mainly as immunosuppressive virulence factors, which could explain the absence of clinical symptoms during STEC infections [10]. Toxins secreted by bovine intestinal STEC have been shown to act primarily on intraepithelial lymphocytes, affecting the host immune response and the correct development of intestinal epithelial cells [10,38,39]. Additionally, the hemolysins, which increase the capacity to take up iron, and the intimin, fundamental for the induction of A/E lesions, may participate in the processes of colonization, immunosuppression and infection [10,39,40]. These observations lead us to believe that intestinal *E. coli* harboring the *stx1*/*eae*/*ehxA* genotype hold a role in calves’ mortality.

The most prevalent gene among the dead animals was *iucD* (82%, n = 17), which could indicate the importance of this siderophore for the survival of *E. coli*. This prevalence was similar to that previously reported in live calves with and without symptoms of NCD in Uruguay, although significantly higher than that reported in live animals from dairy farms in Argentina (OR = 10.73; 95% CI, 5.50–20.93) [26,27]. Additionally, the prevalence of *papC* and *afa8E* adhesins genes were 80% and 50%, respectively, with both significantly more represented in dead than live calves in Uruguay (OR = 2.59; 95% CI, 1.49–4.59, OR = 1.89, 95% CI, 1.08–3.32, respectively) [26] and Argentina (OR = 3.31; 95% CI, 1.85–5.93, OR = 7.53; 95% CI, 3.73–15.20, respectively) [27]. The F17 and CS31A genes had high prevalence, like those previously observed in other studies [26,27]. The *f17GI* variant was more represented in dead than live animals in previous works (OR = 6.70; 95% CI, 1.93–23.20), suggesting a role for the adhesin F17GI variant in mortality. On the other hand, *f17GII* had a lower prevalence in dead calves (OR = 0.48; 95% CI, 0.25–0.93), suggesting a role in the intestines of living calves [26]. Only one animal had EPEC isolates (*eae*+). Again, this result coincides with that reported in live animals, suggesting that NTEC, EPEC and ETEC are not relevant in the mortality of Uruguayan dairy calves [26,27].

Subsequently, unique isolates with STEC, EPEC and NTEC virulence profiles were selected in each biological sample (n = 20). Among these, the occurrence of isolates with STEC virotype was very high (85%). In agreement with previous works [25], the 16 STEC isolates were identified with *stx1*a and c subtypes, confirming their circulation in dead calves from Uruguay. These isolates were mainly assigned to groups B1 and E, followed by groups C and A, whereas the two hybrid STEC/NTEC isolates were assigned to the phylogroup A. The NTEC isolates were also assigned to phylogroup B1 and the EPEC to phylogroup E. Our results only partially coincide with those of the literature, where STEC strains are often assigned to phylogroups B1 and E and EPEC to E [41]. In Uruguay, the heterogeneity in phylogenetic groups among STEC was previously noted in isolates recovered from bovine feces [25].

Like human *E. coli* strains, bovine strains belong to a continuously increasing number of new serogroups, many of which are also associated with human disease. Despite this, O157:H7 EHEC has been exceptionally associated with diarrhea in young calves [6]. In this work, nine STEC isolates were assigned to serogroup O111, whereas a high number of strains were not typeable to any of the serogroups tested, suggesting the circulation of serogroups other than O157:H7 and the “big six”. The results partially coincide with those previously reported in Uruguay for calves [25], where serogroups O103 and O111 were detected. Furthermore, O111 STEC isolates have been associated with severe disease in children in the country [42].

All the isolates were AMP resistant (20/20), followed by STR and NAL (19/20 and 17/20, respectively). The resistance percentages to CIP and ENR were 80% (16/20) and 65% (13/20). The CN, STX, TOB and AMC resistance was 30% (6/20), 25% (5/20), 20% (4/20) and 15% (3/20), respectively (Figure 3). The isolates presented different antibiotic profiles and almost all were multidrug-resistant (16/20), including 13 of the 17 selected STEC. Other studies have indicated a possible higher presence of antibiotic resistance among non-O157 STEC serotypes from animal origin, including resistance against antimicrobials that are critical for human and veterinary medicine, carrying *eae* and *stx1* genes, a combination of virulence and serotype detected in our work [43,44]. The high percentage of MDR-STEC isolates detected in our study differs from what was previously observed in STEC from feces of live calves [25]. In both studies, resistance to Ampicillin was the most prevalent among STEC, consistent with the fact that β-lactams are the most widely used antibiotics in animals [45]. However, no non-O157 MDR-STEC strains were identified in living calves [25]. This difference could reflect the extensive use of antimicrobials in dairy herds to treat infectious diseases such as NCD, particularly in animals with a high risk of dying. Additionally, recent work carried out in isolates from different sources including human samples and children’s feces, beef and carcasses in our country did not identify a high level of antimicrobial resistance in STEC [46,47]. In our work, antimicrobial resistance and even multi-resistance seems alarming. This difference in the results could suggest that multi-resistance to antimicrobials could be related to the treatment of calves in severe enteric disease. New work is required to demonstrate this association. The presence of *qnrB19* was detected. This genetic variant has previously been identified in animals and humans in our country [24,48,49]. Although it is accepted that these determinants do not confer a high level of resistance, they can potentiate other mechanisms and spread easily, representing the emergence of new variants of MDR-non O157 STEC [50].

The PFGE analysis showed a high diversity among the isolates recovered from dead animals, even considering the isolates recovered from live animals included as controls. Isolates from the same animal and herd presented more similar patterns to those from other animals, and no similarity was observed between intestinal isolates and extraintestinal isolates. Intestinal and extraintestinal STEC isolates from the same animal were grouped into the same variant, even in the same clone. This result suggests that intestinal STEC may have reached other organs when the disease worsened. Isolates from the same clone presented different virulence profiles and antimicrobial resistance profiles. This observation highlights the mobility of the genes that encode them.

## 5. Conclusions

In Uruguay, neonatal calf mortality is high and a concern for farmers. This study was the first in Uruguay that aimed to characterize pathogenic *E. coli* in dairy calves killed by enteric disease. Most of the isolates were characterized as EHEC (*stx1*+, *eae*+, *ehxA*+), and a minority corresponded to STEC LEE− (*stx1*+, *eae*−), NTEC (*cnf1*+) and EPEC (*eae*+, *stx*−) virotypes. The high presence of EHEC in the intestines of dead animals differed from what has been previously observed in live calves in our country. Therefore, it can be speculated that the presence of EHEC in the intestines of calves could be a risk factor for mortality and its detection could indicate a bad prognosis. Likewise, *afa8E*, *papC*, *f17GI* and *f17GII* were also found in different prevalence compared with live animals in our country, which is highly relevant when designing mortality-prevention and -control strategies. A high prevalence of MDR, particularly MDR-STEC, was observed in pathogenic *E. coli* isolates from dead calves, and the circulation of *qnr* genes was evidenced. It is essential to take action to minimize the emergence and spread of resistance between animals, the environment and humans.

## Figures and Tables

**Figure 1 microorganisms-11-01704-f001:**
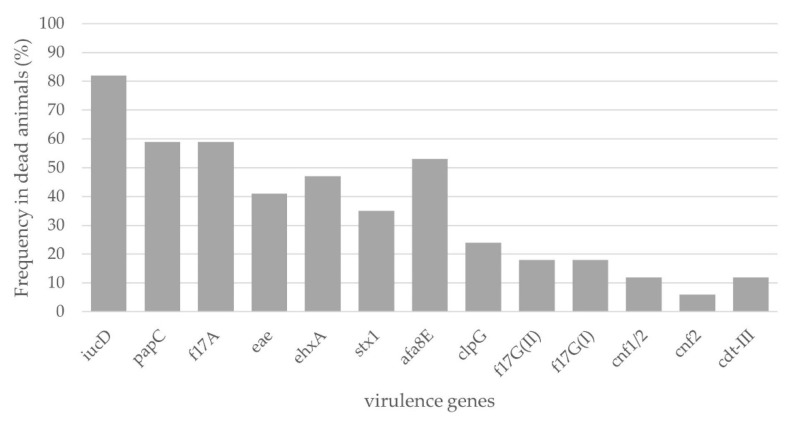
Frequency of occurrence of *E. coli* virulence genes in dead animals (n = 17).

**Figure 2 microorganisms-11-01704-f002:**
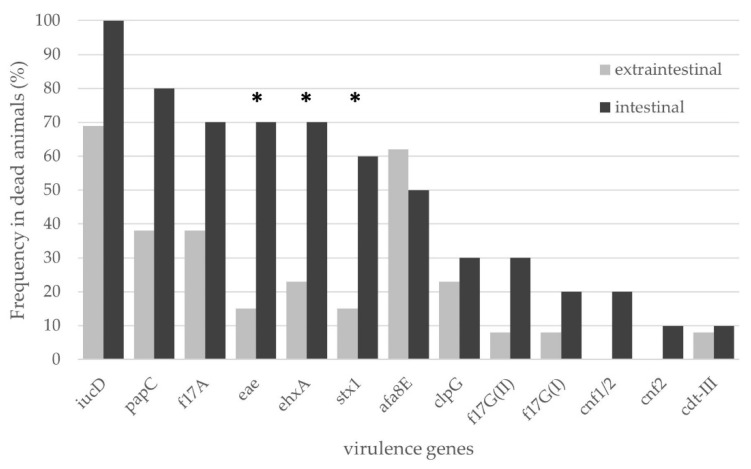
Comparison of virulence gene frequencies in the intestinal and extraintestinal environment of dead animals. * Indicates genes with significant differences between the different samples.

**Figure 3 microorganisms-11-01704-f003:**
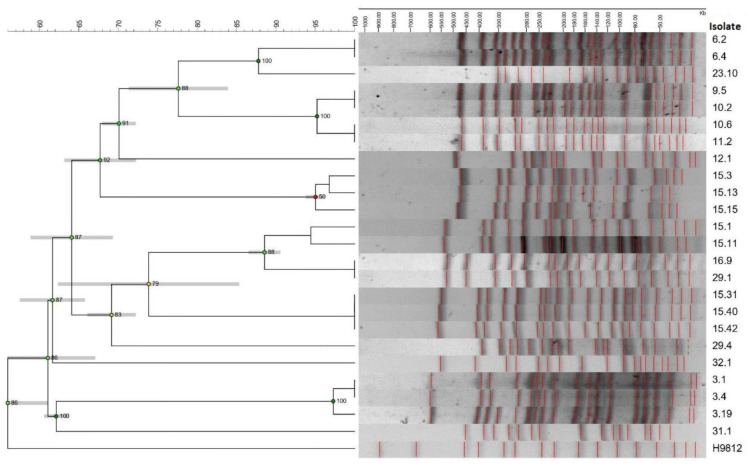
Clonal relationship of STEC/EPEC/NTEC mortality isolates. Dendrogram made with the GelCompar II program (Applied Maths, Version 6.5), UPGMA method, Dice coefficient, 1% optimization and 1% tolerance. H9812: reference standard *Salmonella enterica* subsp. *enterica* serovar Braenderup. Five *E. coli* isolates from feces of asymptomatic calves living with dead calves on dairy farms K and M (isolates 23.10 and 29.1, 29.4, 31.1 and 32.1, respectively) were incorporated into the analyses.

**Table 1 microorganisms-11-01704-t001:** Origin, age and date of death of the calves.

Animal	Establishment	Department	Date	Age
1	A	Colonia	July-2015	w/d
2	B	Colonia	July-2015	w/d
3	C	Colonia	December-2015	w/d
4	D	Río Negro	May-2016	5 days
5	E	San José	October-2016	15 days
6	F	Río Negro	April-2017	w/d
7	G	Colonia	July-2017	17 days
8	H	Colonia	July-2017	2 years
9	I	San José	August-2017	5–12 days
10	I	San José	August-2017	5–12 days
11	I	San José	August-2017	5–12 days
12	J	Colonia	August-2017	30 days
13	J	Colonia	September-2017	2 days
14	K	Colonia	September-2017	9 days
15	L	Colonia	October-2017	15 days
16	M	Colonia	October-2017	10 days
17	N	Canelones	November-2017	w/d

w/d: without data.

**Table 2 microorganisms-11-01704-t002:** Origin and virulence profile of selected STEC/EPEC/NTEC isolates. Isolates beginning with the same number correspond to the same animal.

Isolate	Origin	Virulence Profile	Virotype	Phylogroup	Subtype of Stx1	Serogroup
3.1	feces	*eae*/*stx1*/*ehxA*/*cnf1*/*iucD*	EHEC/NTEC	A	a and c	n/d
3.4	feces	*eae*/*stx1*/*ehxA*	EHEC	A	a and c	n/d
3.19	feces	*eae*/*stx1*/*cnf1*/*iucD*	EHEC/NTEC	A	a and c	n/d
6.2	feces	*eae/ehxA/iucD*	EPEC	E	n/a	n/d
6.4	feces	*eae/ehxA*	EPEC	E	n/a	n/d
9.5	intestine	*eae*/*stx1*/*ehxA*/*iucD*	EHEC	E	a and c	n/d
10.2	intestine	*eae*/*stx1*/*ehxA*/*iucD*	EHEC	E	a and c	n/d
10.4	lung	*eae*/*stx1*/*ehxA*/*iucD*	EHEC	B1	a and c	n/d
10.6	MLN *	*eae*/*stx1*/*ehxA*/*iucD*	EHEC	E	a and c	n/d
11.2	feces	*eae*/*stx1*/*ehxA*/*iucD*	EHEC	E	a and c	n/d
12.1	feces	*f17A*/*f17GII*/*cnf1*/*cnf2*/*cdtIII*/*iucD*	NTEC	B1	n/a	n/d
15.1	feces	*eae*/*stx1*/*ehxA*/*iucD*	EHEC	B1	a and c	O111
15.3	feces	*eae*/*stx1*/*ehxA*/*iucD*/*afa8E*	EHEC	C	a and c	O111
15.11	brain	*eae*/*stx1*/*ehxA*/*iucD*	EHEC	B1	a and c	O111
15.13	brain	*eae*/*stx1*/*ehxA*/*iucD*/*afa8E*	EHEC	C	a and c	O111
15.15	brain	*eae*/*stx1*/*iucD*/*afa8E*	EHEC	C	a and c	O111
15.31	liver	*eae*/*stx1*/*ehxA*/*iucD*	EHEC	B1	a and c	O111
15.40	lung	*eae*/*stx1*/*ehxA*/*iucD*	EHEC	B1	n/d	O111
15.42	lung	*stx1*/*ehxA*/*iucD*	STEC LEE−	B1	a and c	O111
16.9	feces	*eae*/*stx1*/*ehxA*	EHEC	B1	a and c	O111

* MLN: mesenteric lymph node; n/d: undetermined; n/a: does not apply.

**Table 3 microorganisms-11-01704-t003:** Origin and resistance profile of selected STEC/EPEC/NTEC isolates. Isolates beginning with the same number correspond to the same animal.

Isolate	Origin	Virotype	Resistance Profile	*qnr*
3.1	feces	EHEC/NTEC	AMP/CN/STR	n/a
3.4	feces	EHEC	AMP/CN/TOB/STR	n/a
3.19	feces	EHEC/NTEC	AMP/CN/STR	n/a
6.2	feces	EPEC	AMP/NAL/ENR/CIP/STR/STX	*qnrB19*
6.4	feces	EPEC	AMP/NAL/ENR/CIP/STR/STX	*qnrB*
9.5	intestine	EHEC	AMP/NAL/ENR/CIP/STR	*qnrB19*
10.2	intestine	EHEC	AMP/AMC/CXM/NAL/ENR/CIP/STR	*qnrB19*
10.4	lung	EHEC	AMP/NAL/ENR/CIP	n/d
10.6	MLN *	EHEC	AMP/CXM/NAL/ENR/CIP/STR	*qnrB*
11.2	feces	EHEC	AMP/AMC/CXM/NAL/ENR/CIP/STR	n/d
12.1	feces	NTEC	AMP/NAL/ENR/CIP/STR	*qnrB*
15.1	feces	EHEC	AMP/NAL/CIP/STR	n/d
15.3	feces	EHEC	AMP/AMC/NAL/ENR/CIP/CN/TOB/STR/STX	*qnrB*
15.11	brain	EHEC	AMP/NAL/CIP/STR	*qnrB*
15.13	brain	EHEC	AMP/NAL/ENR/CIP/CN/TOB/STR	*qnrB*
15.15	brain	EHEC	AMP/NAL/ENR/CIP/CN/TOB/STR	n/d
15.31	liver	EHEC	AMP/NAL/CIP/STR	n/d
15.40	lung	EHEC	AMP/NAL/CIP/STR	*qnrB*
15.42	lung	STEC LEE−	AMP/NAL/ENR/CIP/STR/STX	n/d
16.9	feces	EHEC	AMP/NAL/ENR/CIP/STR/STX	*qnrB*

* MLN: mesenteric lymph node; n/d: undetermined; n/a: does not apply.

## Data Availability

No new data were created or analyzed in this study. Data sharing is not applicable to this article.

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
