# Peer review of "Shiga Toxin-Producing Escherichia coli (STEC) Associated with Calf Mortality in Uruguay"

_microorganisms, 2023, doi:10.3390/microorganisms11071704_

Round 1

Reviewer 1 Report

Abstract

Clarify these abbreviation “Genes characteristic of the STEC, EPEC and NTEC were identified”

Introduction

Authors should write the whole word in first time, there are many abbreviations in introduction without description like LEE

Materials and methods

Authors should determine the type of collected samples

The language is good

Author Response

Comments to the reviewers:

#1 Comments and Suggestions for Authors

Abstract
Clarify these abbreviation “Genes characteristic of the STEC, EPEC and NTEC were identified”. The virotypes were clarified in the abstract as requested.

Introduction
Authors should write the whole word in first time, there are many abbreviations in introduction without description like LEE. The description of the abbreviations were incorporated as requested.

Materials and methods
Authors should determine the type of collected samples. The type of collected samples is presented in supplementary Table S2.

Reviewer 2 Report

This scientific paper provides a comprehensive analysis of the issues surrounding Shiga toxin-producing Escherichia coli (STEC) and antimicrobial resistance (AMR) in neonatal calves. The authors have presented a well-researched and soundly written account of the problematics associated with these two critical areas of concern. The introduction provided a comprehensive overview of the current state of knowledge, while the materials and methods utilized were both adequate and contemporary in order to derive the conclusions presented. The discussion was adequately expanded upon.

Author Response

Reviewer #2 left no suggestions for improving the manuscript.

Reviewer 3 Report

The work of Umpierrez et al details the isolation and characterisation of STEC strains from dead newborn calves. The study is warranted by the need to better understand the co-existence of these pathogens and cattle, as well as to explain their potential etiological role in neonatal calf diarrhea. They have found a high prevalence of STEC and EHEC, with EHEC / NTEC hybrid pathotypes as well. Many of the strains proved to be multidrug resistant as well.

The study could have been enriched with sequencing strains representing the most virulent / interesting virulence and/or resistance gene set. Information about the antimicrobial susceptibility of the strains could have been enriched by performing MIC tests as well.

Nevertheless, the work is valuable, and was performed with state-of-the-art methods. It is well-structured and well-written, with the Introduction and Discussion providing adequate overview of recent results of the field.

As a general comment, I find the use of LEE+ STEC as a pathotype designation a little bit awkward, as we already have a proper designation for these strains: EHEC. I strongly suggest to use it, as it would be more comprehensible and would attract more readers if used in the article, as an additional keyword for example.

It is understandable that the authors only included strains with unique virulence profiles in the tables. However, it would have been more elegant to include all the virulence and resistance profiles of all strains in a supplementary table, and attach it to the manuscript as well.

A few additional minor suggestions:

The lines are not numbered, this made it hard for me to reference the suggested changes.

Introduction, page 2:

DIEA, 2020’ I cannot find this reference in the list.

Materials and methods

For the serogroup-specific and phylogenetic groups PCR, as well as the stx subtyping and virulence gene detection PCRs, the authors refer to their own earlie works - however, in these papers, the authors used reactions which had been published much earlier. Please refer to these works properly (Paddock et al, 2012, for the serogroup-specific PCR, Clermont et al, 2013 for the phylogenetic grouping, and Scheutz et al, 2012 for stx subtyping, for example).

The authors also investigated the cdt-III gene cluster, of which the active subunit is essentially the same as that of cdt-V. Did the authors perhaps use another reaction to differentiate? cdt-V is a frequent additive virulence gene of EHEC strains, so it is possible that in their case it was cdt-V, and not cdt-III.

Section 3.2

The abbreviation ‘MLN’ occurs here first, but is not explained, only in a note attached to Table 2 later. Please introduce it at its first occurrence.

Section 3.3.1 I suggest ‘investigated’ instead of ‘studied’

Section 3.6 ‘typed’ instead of ‘typified’

Author Response

#3 Comments and Suggestions for Authors

The work of Umpierrez et al details the isolation and characterisation of STEC strains from dead newborn calves. The study is warranted by the need to better understand the co-existence of these pathogens and cattle, as well as to explain their potential etiological role in neonatal calf diarrhea. They have found a high prevalence of STEC and EHEC, with EHEC / NTEC hybrid pathotypes as well. Many of the strains proved to be multidrug resistant as well.
The study could have been enriched with sequencing strains representing the most virulent / interesting virulence and/or resistance gene set. Information about the antimicrobial susceptibility of the strains could have been enriched by performing MIC tests as well.
Nevertheless, the work is valuable, and was performed with state-of-the-art methods. It is well-structured and well-written, with the Introduction and Discussion providing adequate overview of recent results of the field.

As a general comment, I find the use of LEE+ STEC as a pathotype designation a little bit awkward, as we already have a proper designation for these strains: EHEC. I strongly suggest to use it, as it would be more comprehensible and would attract more readers if used in the article, as an additional keyword for example. The suggestion was accepted and the word “EHEC” was incorporated along the document when refering to LEE+ STEC isolates.

It is understandable that the authors only included strains with unique virulence profiles in the tables. However, it would have been more elegant to include all the virulence and resistance profiles of all strains in a supplementary table, and attach it to the manuscript as well. The large table with all the isolates was incorporated as supplementary Table S2.

 A few additional minor suggestions:
The lines are not numbered, this made it hard for me to reference the suggested changes. The lines now are numbered.

Introduction, page 2:
‘DIEA, 2020’ I cannot find this reference in the list. The reference was included in the reference list.

Materials and methods

For the serogroup-specific and phylogenetic groups PCR, as well as the stx subtyping and virulence gene detection PCRs, the authors refer to their own earlie works - however, in these papers, the authors used reactions which had been published much earlier. Please refer to these works properly (Paddock et al, 2012, for the serogroup-specific PCR, Clermont et al, 2013 for the phylogenetic grouping, and Scheutz et al, 2012 for stx subtyping, for example). The references were changed as requested.

The authors also investigated the cdt-III gene cluster, of which the active subunit is essentially the same as that of cdt-V. Did the authors perhaps use another reaction to differentiate? cdt-V is a frequent additive virulence gene of EHEC strains, so it is possible that in their case it was cdt-V, and not cdt-III. Although seems interesting, we did not investigate the cdt-V gene cluster in this work. We will consider it for future works.

Section 3.2
The abbreviation ‘MLN’ occurs here first, but is not explained, only in a note attached to Table 2 later. Please introduce it at its first occurrence.The abbreviation was explained as requested.
Section 3.3.1

I suggest ‘investigated’ instead of ‘studied’. The word was replaced as suggested.
Section 3.6

‘typed’ instead of ‘typified’ The grammar mistake was corrected.
